# Comparison of CD3e Antibody and CD3e-sZAP Immunotoxin Treatment in Mice Identifies sZAP as the Main Driver of Vascular Leakage

**DOI:** 10.3390/biomedicines10061221

**Published:** 2022-05-24

**Authors:** Shihyoung Kim, Rajni Kant Shukla, Eunsoo Kim, Sophie G. Cressman, Hannah Yu, Alice Baek, Hyewon Choi, Alan Kim, Amit Sharma, Zhirui Wang, Christene A. Huang, John C. Reneau, Prosper N. Boyaka, Namal P. M. Liyanage, Sanggu Kim

**Affiliations:** 1Department of Veterinary Biosciences, The Ohio State University, Columbus, OH 43210, USA; kim.6754@buckeyemail.osu.edu (S.K.); shukla.93@osu.edu (R.K.S.); eunsoo_kim@pusan.ac.kr (E.K.); cressman.9@buckeyemail.osu.edu (S.G.C.); yu.2127@osu.edu (H.Y.); baek.71@osu.edu (A.B.); choi.1502@osu.edu (H.C.); kim.6396@buckeyemail.osu.edu (A.K.); sharma.157@osu.edu (A.S.); boyaka.1@osu.edu (P.N.B.); namal.malimbadaliyanage@osumc.edu (N.P.M.L.); 2Department of Microbial Immunity and Infection, The Ohio State University, Columbus, OH 43210, USA; 3Infectious Diseases Institute, The Ohio State University, Columbus, OH 43210, USA; 4Department of Surgery, University of Colorado Denver Anschutz Medical Campus, Division of Plastic & Reconstructive Surgery, 12700 East 19th Avenue, Aurora, CO 80045, USA; zhirui.wang@cuanschutz.edu (Z.W.); christene.huang@cuanschutz.edu (C.A.H.); 5Department of Surgery, University of Colorado Denver Anschutz Medical Campus, Division of Transplant Surgery, 12700 East 19th Avenue, Aurora, CO 80045, USA; 6Division of Hematology, The Ohio State University, Columbus, OH 43210, USA; john.reneau@osumc.edu

**Keywords:** CD3e monoclonal antibody, CD3e immunotoxin, T-cell depletion, saporin, saporin–streptavidin (sZAP), vascular leakage syndrome, polymorphonuclear leukocytes (PMN), intravascular staining, mouse model, immunotherapy

## Abstract

Anti-CD3-epsilon (CD3e) monoclonal antibodies (mAbs) and CD3e immunotoxins (ITs) are promising targeted therapy options for various T-cell disorders. Despite significant advances in mAb and IT engineering, vascular leakage syndrome (VLS) remains a major dose-limiting toxicity for ITs and has been poorly characterized for recent “engineered” mAbs. This study undertakes a direct comparison of non-mitogenic CD3e-mAb (145-2C11 with Fc-silent^TM^ murine IgG1: S-CD3e-mAb) and a new murine-version CD3e-IT (saporin–streptavidin (sZAP) conjugated with S-CD3e-mAb: S-CD3e-IT) and identifies their distinct toxicity profiles in mice. As expected, the two agents showed different modes of action on T cells, with S-CD3e-mAb inducing nearly complete modulation of CD3e on the cell surface, while S-CD3e-IT depleted the cells. S-CD3e-IT significantly increased the infiltration of polymorphonuclear leukocytes (PMNs) into the tissue parenchyma of the spleen and lungs, a sign of increased vascular permeability. By contrast, S-CD3e-mAbs-treated mice showed no notable signs of vascular leakage. Treatment with control ITs (sZAP conjugated with Fc-silent isotype antibodies) induced significant vascular leakage without causing T-cell deaths. These results demonstrate that the toxin portion of S-CD3e-IT, not the CD3e-binding portion (S-CD3e-mAb), is the main driver of vascular leakage, thus clarifying the molecular target for improving safety profiles in CD3e-IT therapy.

## 1. Introduction

Immunotoxins (ITs) and antibody therapies have been developed to treat various diseases, including cancers, infectious diseases, autoimmune diseases, and organ transplant [1,2,3,4,5,6,7,8,9,10]. A major side effect limiting the clinical application of ITs and some antibody therapies has been vascular leak syndrome (VLS) [8,11,12,13]. While symptoms are highly variable among patients, VLS can be characterized by the extravasation of fluids, plasma proteins, and polymorphonuclear leukocytes (PMNs) due to an increase in vascular permeability, resulting in interstitial edema and organ failure [14,15]. The etiology of these adverse effects is complex and remains incompletely understood. Studies have shown that the permeability of the blood vessels is largely governed by mechanisms that control endothelial barrier functions for organ homeostasis. Increased vascular permeability is a prominent feature of local inflammation, innate immunity, and tissue damage. Mediators such as histamine, bradykinin, PMN chemoattractant, vascular endothelial growth factor, tumor necrosis factor, and various cytokines released from infiltrated PMNs, and dying cells have been shown to increase the permeability of endothelial cell (EC) junctions [16,17,18,19].

First-generation antibody therapeutics, including anti-human CD3 mAbs (OKT3), have been shown to activate large numbers of immune cells to release several cytokines, leading to a set of side effects, including VLS and cytokine release syndrome [20,21,22,23,24,25]. Activated T cells may directly interact with and damage endothelial cells [13,26]. Similarly, first-generation CD3 ITs—for example, a mutant DT (CRM9) conjugated with anti-rhesus CD3e (FN18 with murine IgG1)—demonstrated severe side effects including proinflammatory cytokine release and VLS, which compromised the treatment efficacy and led to the death of some monkeys [27,28]. Unlike these first-generation therapeutic agents in the 1990s, most of which involve foreign antibodies, recent recombinant antibodies and ITs—that were “humanized” or removed of the foreign portion (e.g., murine IgG) and the Fc receptor binding ability of the mAbs—have demonstrated significantly improved treatment efficacy and safety profiles [29,30,31,32]. The “second-generation” recombinant CD3e ITs (e.g., Resimmune and A-dmDT390-scfbDb(C207)) tested in recent preclinical and clinical studies are structurally much simpler: they are composed of a truncated DT (DT390: the catalytic and translocation domains of DT) fused to two single-chain antibody fragments reactive with CD3e [33,34]. Resimmune (A-dmDT390-bisFv(UCHT1)) has demonstrated significantly improved safety profiles compared to the first-generation CD3e IT. Unlike the first-generation OKT3 or FN18-CRM9, the second-generation CD3e IT did not notably increase cytokine levels [35,36]. Although less immunogenic, Resimmune seemed to continue to cause VLS with relatively mild symptoms. Of 30 patients treated for T-cell lymphomas with Resimmune, 10 patients had grade 2 VLS, and 2 with a history of heart disease had grade 4–5 VLS [2]. The relationship and relative contributions of different portions of CD3e IT to VLS induction remain unclear.

The potential mechanisms underlying IT-induced VLS include (i) endothelial cell damage by direct interactions with the antibody or toxin portion [37,38,39,40], (ii) local inflammation induced by toxin-mediated cell death [41,42,43], and/or (iii) excessive cytokine release by antibody-activated leukocytes [20,21,22,23,24,25]. VLS-inducing IL-2, bacteria toxins, and ribosomal inactivating proteins are known to harbor a common (*x*)D(*y*) structural motif (*x* = L, I, G, or V and *y* = V, L, or S) that mediates its binding to endothelial cells, leading to endothelial cell damage and VLS [44]. The (*x*)D(*y*) motif-mediated endothelial cell damage does not influence the enzymatic activity of the toxin, and the removal or mutation of these motifs significantly reduces VLS [38,39]. Once internalized into the target cells, the catalytic activity of the toxins or ribosomal inactivating proteins induces cell death, which may increase the circulation of pro-inflammatory cytokines and chemokines that can also induce endothelial cell leakage [43,45,46]. Some IT studies, however, have only shown an insignificant increase in the circulation of inflammatory cytokines [35,36,47].

Here, we dissected and compared the roles of the different components of CD3e IT (the CD3e binding portion and toxin portion) in inducing PMN infiltration into local organs. Inspired by the advanced intravascular staining methods that can track and quantify the migration of leukocytes both in the vasculature and in the tissue parenchyma [48,49,50,51], we generated a new murine VLS study model to evaluate these agents in the lungs. We created a new murine CD3e IT (S-CD3e-IT) using the streptavidin–saporin (sZAP) conjugation system and analyzed saporin-induced VLS as a model protein toxin. Saporin toxins have been increasingly used in recent in vivo animal studies [52,53]. However, to date, the causal effects of saporin on VLS remain poorly investigated and somewhat controversial [54,55,56]. We used a new chimeric, non-mitogenic CD3e-mAb (145-2C11 with mouse IgG1 with Fc-Silent^TM^ mutations, Absolute Antibody, denoted as S-CD3e-mAb) to evaluate the roles of the CD3e-binding portion of IT. We demonstrate significantly improved safety profiles of S-CD3e-mAb compared to wild-type hamster 145-2C11 (W-CD3e-mAb). W-CD3e-mAb that has been commonly used in vivo studies can induce unwanted side effects due to its foreign immunoglobulin [22,30,31,57]. New CD3e-saporin ITs (S-CD3e-IT) were well-tolerated in mice. Both the S-CD3e-mAb and S-CD3e-IT demonstrated distinct, but expected modes of action specific to T cells. S-CD3e-IT showed significant signs of vascular leakage, whereas S-CD3e-mAb had an insignificant impact. We also found that a control IT (sZAP-conjugated with an isotype mAb with Fc Silent murine IgG1, Absolute Antibody) that does not exhibit T-cell depletion can still cause vascular leakage, indicating the importance of the saporin molecule itself in inducing VLS in mice. S-CD3e-IT and DT treatments demonstrated a comparable level of vascular leakage in transgenic mice that express DT receptors on T cells (CD4iDTR mice).

## 2. Materials and Methods

### 2.1. Mice

C57BL/6J, and the breeding pairs of B6.Cg-Tg(Cd4-are)1Cwi/BfluJ and C57BL/6-Gt(Rosa)26Sor^tm1(HBEGF)Awai^/J were purchased from the Jackson Laboratory (Bar Harbor, ME, USA). F1 CD4iDTR mice were generated by crossing B6.Cg-Tg(Cd4-are)1Cwi/BfluJ and C57BL/6-Gt(Rosa)26Sor^tm1(HBEGF)Awai^/J as described previously [58]. The DNA for genotyping was prepared from ear punching. Genotyping was confirmed by PCR using the following primer pairs: for CD4-cre (336 bp product), 5′-gttctttgtatatattgaatgttagcc-3′ (Common Forward), 5′-tatgctctaaggacaagaattgaca-3′ (Wild-type Reverse), and 5′-ctttgcagagggctaacagc-3′ (Mutant Reverse); and for iDTR (340 bp product), 5′-gcgaagagtttgtcctcaacc-3′ (Mutant Reverse), 5′-aaagtcgctctgagttgttat-3′ (Common Forward), and 5′-ggagcgggagaaatggatatg-3′ (Reverse). All mice in this study were maintained in accordance with the guidelines of Ohio State University (OSU). All animal experiments were conducted with the protocol approved by the OSU Animal Care and Use Committee.

### 2.2. Biotinylation of Antibodies and S-CD3e-IT Preparation

CD3e monoclonal antibodies (145-2C11 clone) with Fc-Silent mouse IgG1 were purchased from Absolute Antibody. Biotinylation of Fc-silent CD3e monoclonal antibody (S-CD3e-mAb) was carried out using FluoReporter Mini-Biotin-XX Protein Labeling Kit from Invitrogen and according to the recommended manufacturer manual. An optimal ratio of company-provided biotin solution (μL): IgG (mL; 1 mg/mL concentration) was chosen to be 4:1 based on 3–8% Tris–acetate gradient gel analysis (Invitrogen, Waltham, MA, USA). Anti-murine-CD3e immunotoxin (S-CD3e-IT) was prepared by conjugating biotinylated-S-CD3e-mAb with streptavidin–ZAP (Advanced Targeting System, San Diego, CA, USA) in a 1:1 molar ratio. Gel images are shown in Figure 1B. The molar ratios were calculated based on the company-provided molecular weight information. S-iso-IT was generated by conjugating isotype-matched antibody with Fc-Silent mutant (Absolute Antibody, Upper Heyford, UK) with sZAP with similar procedures described above.

### 2.3. S-CD3e-IT In Vitro Cell Killing Assay

In vitro cell-killing experiments were conducted using C57BL/6J spleen cells. CD3+ T cells were negatively selected by MojoSort Mouse CD3 T cell Isolation Kit (BioLegend, San Diego, CA, USA). CD3+ cells were incubated with various concentrations of S-CD3e-IT (1 pM to 10 nM) for 30 min in a complete RPMI medium (Hyclone, Logan, UT, USA) supplemented with 10% Fetal Bovine Serum (Hyclone, Logan, USA), 0.1% 2-mercaptoethanol (Gibco, Waltham, MA, USA), 1% L-Glutamine, and 1% antibiotic cocktail in 37 °C and 5% CO_2_ incubator. After washing the cells twice, the cells were plated in flat-bottom 96-well plates (Falcon, Tewksbury, MA, USA) with plate-bound 5 μg/mL anti-CD3 (145-2C11, BioLegend) and soluble 2 μg/mL anti-CD28 (37.51, BioLegend, San Diego, CA, USA) in a complete RPMI medium supplemented as above. After 72 h, T subset viability was analyzed by Cytek Aurora flow cytometry (Cytek Biosciences, Fremont, CA, USA) using previously published protocol with modifications [59,60]. The cell viability at different concentrations of S-CD3e-IT was normalized with that of the no-treatment (PBS) control.

### 2.4. In Vivo Treatment with CD3e Monoclonal Antibody and S-CD3e-IT

CD3e monoclonal antibodies (145-2C11) with hamster IgG1 (W-CD3e-mAb) and chimeric 145-2C11 with Fc-Silent murine IgG1 (S-CD3e-mAb) were purchased from Bio x Cell and Absolute Antibody, respectively. For CD3e mAb treatment, C57BL/6J mice were treated with 40 μg CD3e mAb in 200 μL sterile PBS via single retro-orbital injection per day for five days and were euthanized on day 8, as described previously [61,62]. For S-CD3e-IT treatment, C57BL/6J mice received 25 μg S-CD3e-IT in sterile 200 μL PBS twice a day via retro-orbital injection for four days and were euthanized on day 6. For S-iso-IT treatment, C57BL/6J mice were treated with 15 μg S-iso-IT in sterile 200 μL PBS twice a day via retro-orbital injection for four days and were euthanized on day 6. There was no notable difference in leukocyte composition between 15 μg and 25 μg S-iso-IT-treated animals in our preliminary tests. For CD4-iDTR mouse tests, we treated CD4-iDTR mice with 200 ng diphtheria toxin (Sigma, St. Louis, MO, USA) in sterile 200 μL water twice a day for four days via retro-orbital injection, and euthanized the mice on day 5. The S-CD3e-IT treatment strategy was the same as above.

### 2.5. Intravascular Staining and Cell Isolations

On the day of euthanasia, the mice received a total of 3 mg of PE/Cy7-CD45.2 (104, BioLegend) in 200 μL sterile DPBS via retro-orbital injection. After 3 min of injection, the mice were euthanized following the previously established protocols [48,63]. The peripheral blood was collected from the heart. The spleen, mesenteric LN, lung, and bone marrow (right femur) were harvested, and tissues were processed to isolate leukocytes following previous studies with modifications [64,65].

### 2.6. Flow Cytometry

The collected leukocytes from peripheral blood, mesenteric LN, lung, and bone marrow were incubated with Fc-blocking TruStain FcX^TM^ (anti-mouse CD16/32, BioLegend, San Diego, CA, USA) antibody, followed by Zombie aqua (live/dead indicator, BioLegend, San Diego, CA, USA). The cells were stained with fluorescence-labeled antibodies. The following antibodies were purchased from BioLegend (San Diego, CA, USA): BV650-CD45.2 (104), Alexa Flour 700-CD3 (500A2), Pacific Blue-CD4 (RM4-5), BV570-CD8 (53-6.7), BV711-CD19 (6D5), and FITC-Gr-1 (RB6-8C5). All samples were acquired on a Cytek Aurora flow cytometry (Cytek Biosciences, Fremont, CA, USA), and the data were analyzed with FlowJo software (BD Bioscience, Franklin Lakes, NJ, USA). The calculation of the absolute number of leukocytes was carried out using CountBright^TM^ Absolute Counting Beads (Invitrogen, Waltham, USA) and according to the recommended manufacturer manual, with modifications. To calculate the absolute count, the following equation was applied:$$Absolute\;count\;\left(cells\right)=\frac{\left(Number\;of\;cell\;events\times assigned\;beads\;count\;of\;the\;lot\right)}{Number\;of\;bead\;events}\times \frac{1000\;µL}{The\;vol\;of\;2\times 10^{6}\;cells}$$

### 2.7. PMN Mobilization and W/N Ratios

Lymphocytes and polymorphonuclear leukocytes (PMN) were identified using the flow cytometry gating strategy shown in below. To calculate PMN mobilization, the absolute number of lymphocytes and PMN was divided by that of PBS for normalization. For evaluating the ratios of organ weight and total cell count (W/N ratios) recovered from the organ, organ weight was measured on the day of euthanasia. The organ total cell count was measured using a Luna-FL^TM^ Dual Fluorescence Cells Counter (Logos Biosystems, Anyang-si, Korea). The organ weight was divided by the total cell count of the organ to evaluate W/N ratios. The W/N ratio of each reagent-treated mouse was divided by that of PBS for normalization.

### 2.8. In Vivo Short-Term Cell-Binding Assay

C57BL/6J mice received a single injection of 15 μg S-CD3e-IT or S-iso-IT retro-orbitally, followed by PE/Cy7-CD45.2 injection via retro-orbital injection for intravascular staining in order to separately analyze leukocytes in the vasculature and in the tissue parenchyma. After 3 min of injection, the mice were euthanized, and peripheral blood and spleen cells were then stained with fluorescence-labeled antibodies including anti-saporin mAb.

### 2.9. Statistics

The results are expressed as mean ± SEM. Statistical significance was evaluated by the Mann–Whitney test. All statistical analyses were conducted with Prism 9 (GraphPad Software, La Jolla, CA, USA).

## 3. Results

### 3.1. Generation of Saporin-Based Anti-Murine CD3epsilon Immunotoxin

CD3e-ITs may induce VLS via (a) the CD3e binding portion of IT, that may induce an excessive activation of T cells, and (b) the toxin portion that can directly damage vascular EC and induce extensive cell death and inflammation. To dissect and compare the roles of these two components of ITs in inducing VLS in mice, we created a new murine CD3e-IT by conjugating the sZAP (Advanced Technology System) with S-CD3e-mAb (Fc silenced, 145-2C11 with mouse IgG1, Absolute Antibody). sZAP is a chemical conjugate of streptavidin and saporin and, thus, can convert biotinylated antibodies into ITs. To create a properly functioning S-CD3e-IT, we optimized both the biotinylation and sZAP conjugation steps (see Figure 1B and the methods section for details). The concentrations of the optimized S-CD3e-IT required to kill 50% of splenic T cells in vitro were comparable to those of previous CD3e-Its [66,67] (Figure 1C). S-CD3e-IT was also effective and specific in depleting T cells in vivo (see Figure 2 below). Notably, a suboptimal IT mix generated with the standard 1:1 ratio of sZAP and biotinylated antibodies (the biotin contents were not optimized for sZAP binding) generated completely misleading results in an in vivo test due to unbound residual antibodies (see Figure 2D).

### 3.2. Non-Mitogenic 145-2C11 (S-CD3e-mAb) and S-CD3e-IT Were Well Tolerated in Mice

For the in vivo analysis, we treated C57BL/6J mice with S-CD3e-IT (25 μg twice daily by retro-orbital injection for four consecutive days). The dosage was chosen to achieve specific and effective (>92%) T-cell depletion in the peripheral blood and spleen based on our preliminary in vivo tests. As a comparison, non-mitogenic chimeric S-CD3e-mAb or native W-CD3e-mAb were also injected retro-orbitally for five consecutive days (40 μg per day). A similar dosage of the W-CD3e-mAb has been shown to effectively modulate immune responses in mice in previous studies [61,68]. One day after the last dosage of S-CD3e-IT and two days after the last dosage of CD3e-mAbs, we euthanized the mice and collected the peripheral blood, spleen, lung, and mesenteric lymph nodes for a pharmacodynamics analysis. Animals treated with W-CD3e-mAb showed moderate weight loss and notable splenomegaly (Figure 2A,B). A comparable level of weight loss and splenomegaly was observed in a previous study using W-CD3e-mAb even at a much lower dosage (5 μg single injection) [22]. On the contrary, non-mitogenic S-CD3e-mAb did not show any sign of weight loss or notable splenomegaly (Figure 2A,B). The S-CD3e-IT-treated animals also demonstrated no notable splenomegaly, but these animals showed transient weight loss (an average of 9–15%; Figure 2A). The body weights quickly returned to their pre-treatment level within 2 to 3 weeks after the treatment (data not shown).

### 3.3. Distinct Modes of Action between S-CD3e-mAb and S-CD3e-IT

S-CD3e-IT-treated mice showed an effective and specific reduction in CD3+ cells in the peripheral blood, spleen, lung, and lymph nodes (Figure 2C, upper panel). S-CD3e-mAb-treated mice also showed a reduction in the number of detectable CD3+ cells, but this was perhaps because the levels of CD3 expression were too low to detect on these cells due to CD3e modulation. CD3e-mAbs were shown to work through CD3e modulation on the surface of T cells, while CD3e IT depleted T cells [69,70,71]. Consistent with these results, S-CD3e-mAb-treated mice showed nearly completely modulated CD3e surface expression on these cells (Figure 2D) while maintaining the total number of CD4+ and CD8+ cells (Figure 2C lower panel). By contrast, S-CD3e-IT treatment resulted in the effective depletion of these cells (Figure 2C, lower panel), with a mild decrease in mean fluorescent intensity (MFI) on the surviving CD4+ and CD8+ cells (Figure 2D). Control W-CD3e-mAb-treated mice showed both CD3e modulation on the cell surface and reduction of CD4+ and CD8+ T-cell numbers in all test organs (except the lungs) (Figure 2C,D). Similar levels of T-cell depletion by W-CD3e-mAb were previously reported [22,61].

The half-life of immunotoxins (including CD3e-IT) are usually only 30 min to several hours [2,72,73]. Although the half-life of S-CD3e-mAb remains unclear, other CD3 antibodies are estimated to be 6 h to 1 day [74,75]. Thus, it is likely that most anti-CD3 mAb and S-CD3-IT had already been cleared in the animal at the time of euthanasia. Flow analysis using anti-TCR flow antibodies also showed no notable differences compared to those using anti-CD3e flow antibodies for S-CD3-IT-treated animal samples (data not shown), indicating negligible competition between flow CD3e antibodies and residual S-CD3e-IT in animal samples during the flow analysis.

### 3.4. S-CD3e-IT Treatment Induces Capillary Leakage

PMN migration from the vasculature to tissue parenchyma is a distinctive feature of local and systemic inflammation and capillary leakage [50,51,76]. We employed the total leukocyte intravascular staining method [48,49] to track PMNs and lymphocytes in the vasculature and in the tissue parenchyma similarly to previous studies, utilizing intravascular Gr-1 staining of PMNs; circulating and tissue-infiltrated PMNs were distinctively identified as Gr-1 positive and Gr-1 negative, respectively [50,51,77]. C57BL/6J mice received the four-day S-CD3e-IT treatment, and on day 6 received intravascular staining antibodies (IVS; anti-CD45.2 with PE/Cy7) by retro-orbital injection three minutes prior to euthanasia (Figure 3A). IVS antibody staining during this short period of time can mark circulating leukocytes in the vasculature (thus IVS+) [48,49,50,51]. After euthanasia, we then stained total organ cells ex vivo with a pool of antibodies that included another anti-CD45.2 (BV650-CD45.2). Circulating lymphocytes and PMNs in the vasculature can thus be identified as IVS+, and those in the tissue parenchyma at the time of injection as IVS− (Figure 3B). In PBS control groups, all leukocytes from the peripheral blood were IVS+ (circulating), while 98% of the lymph node cells were IVS− (tissue resident) as expected. The spleen showed that approximately 80% of the leukocytes are IVS−, and only 20% of the lung cells were IVS−. These results are consistent with previous reports [48,63], indicating expected separation of the circulating and tissue resident leukocytes in different organs in our approach.

PMN migration from the vasculature to tissue parenchyma is a distinctive feature of local and systemic inflammation and capillary leakage [50,51,76]. We employed the total leukocyte intravascular staining method [48,49] to track PMNs and lymphocytes in the vasculature and in the tissue parenchyma similarly to previous studies, utilizing intravascular Gr-1 staining of PMNs; circulating and tissue-infiltrated PMNs were distinctively identified as Gr-1 positive and Gr-1 negative, respectively [50,51,77]. C57BL/6J mice received the four-day S-CD3e-IT treatment, and on day 6 received intravascular staining antibodies (IVS; anti-CD45.2 with PE/Cy7) by retro-orbital injection three minutes prior to euthanasia (Figure 3A). IVS antibody staining during this short period of time can mark circulating leukocytes in the vasculature (thus IVS+) [48,49,50,51]. After euthanasia, we then stained total organ cells ex vivo with a pool of antibodies that included another anti-CD45.2 (BV650-CD45.2). Circulating lymphocytes and PMNs in the vasculature can thus be identified as IVS+, and those in the tissue parenchyma at the time of injection as IVS− (Figure 3B). In PBS control groups, all leukocytes from the peripheral blood were IVS+ (circulating), while 98% of the lymph node cells were IVS− (tissue resident) as expected. The spleen showed that approximately 80% of the leukocytes are IVS−, and only 20% of the lung cells were IVS−. These results are consistent with previous reports [48,63], indicating expected separation of the circulating and tissue resident leukocytes in different organs in our approach.

Compared to the number of tissue-localized (IVS−) PMNs in control mice treated with phosphate-buffered saline (PBS), we found more than two-fold increase in the number of IVS− PMNs in the lungs and spleen of S-CD3e-IT-treated mice (* *p* < 0.05 and ** *p* < 0.01, respectively), indicating increased vascular permeability in these animals (Figure 3C). The number of circulating (IVS+) PMNs was also significantly increased in the spleen (by five-fold; ** *p* < 0.01), reflecting ongoing inflammation. However, there was no significant difference in total lymphocyte populations in both IVS+ and IVS− pools. Unlike PMNs, most lymphocytes (B and T cells) are not required for inducing pulmonary VLS [78].

We further evaluated the ratios of organ weight and total cell count recovered from the organ (W/N ratios) as an estimate for the relative fluid accumulation in the organs (Figure 3D). The S-CD3e-IT group showed a marginal increase in W/N ratios for the lungs compared with those of the PBS control group. There was no notable difference in W/N ratios for the spleen, possibly due to a balanced increase in both the number of total leukocytes and the increase in the spleen weight (shown in Figure 2B). Taken together, these results demonstrate that substantial capillary leakage occurred in the lungs and spleen of S-CD3e-IT-treated animals.

### 3.5. No Notable Sign of Capillary Leakage Following Administration of Non-Mitogenic S-CD3e-mAb

We then tested both S-CD3e-mAb and W-CD3e-mAb for the induction of capillary leakage in comparison with S-CD3e-IT to evaluate the relative impact of the CD3e-binding portion of S-CD3e-IT. Consistent with previous reports on OKT3-induced capillary leakage in humans [20,21,22,23,24,25], mitogenic W-CD3e-mAb-treated mice showed a significant increase in tissue-infiltrated (IVS−) PMNs in both the spleen and lungs (* *p* < 0.05; Figure 3C) and a significant increase in W/N ratios (** *p* < 0.01; Figure 3D) in the spleen, consistent with the significant splenomegaly observed in these mice (shown in Figure 2B). Besides the possible impact of Fc receptor binding by mAbs, the CD3e-binding-mediated T-cell activation can also induce PMN mobilization [79,80]. However, to date, non-mitogenic CD3 mAbs have not been clearly characterized for VLS induction. In our study, non-mitogenic S-CD3e-mAb-treated groups did not show any notable sign of PMN infiltration into the lung and spleen parenchyma (Figure 3C), nor any differences in the W/N ratios compared to PBS controls, which further confirms the low risk of capillary leakage in S-CD3e-mAb-treated mice (Figure 3D). Distinct PMN trafficking and W/N patterns between S-CD3e-mAb and S-CD3e-IT thus suggest the insignificant contribution by the CD3e-binding portion (S-CD3e-mAb) of S-CD3e-IT to the induction of capillary leakage in mice.

### 3.6. Evaluating the Depletion of Local T Cells on IT-Induced Capillary Leakage

Tissue-resident T cells stably reside in the tissue parenchyma and rarely recirculate [81,82,83]. The depletion of these cells in the local tissue by CD3-IT may lead to local inflammation (induced by cell death); nevertheless, the CD3-IT-mediated depletion of these local T cells has not been clearly examined yet, except that skin-resident T cells were shown to be effectively depleted by CD3-IT [84]. We compared S-CD3e-IT and control IT (S-iso-IT) to assess the impact of T-cell death in local tissues on S-CD3e-IT-induced capillary leakage. S-iso-IT was created by conjugating sZAP with an isotype antibody (anti-Fluorescein mouse IgG1 with Fc Silent mutations, Absolute Antibody). First, we conducted a short-term cell-binding assay in vivo to examine (i) whether S-CD3e-IT and S-iso-IT would penetrate the tissue, and (ii) how they bind to target cells. Briefly, C57BL/6J mice received a single injection of S-CD3e-IT or S-iso-IT (15 μg, retro-orbital injection), and five minutes later, they received IVS (PE/Cy7-CD45.2). Three minutes after IVS injection, the mice were euthanized, and their organ cells were then ex vivo stained with an antibody pool that included anti-saporin mAb (Figure 4A). In the S-CD3e-IT-treated group, saporin molecules were bound to >80% of CD4+ cells and >60% of CD8+ cells in the peripheral blood as expected (Figure 4B). In the spleen, saporin was also bound to both IVS+ and IVS− CD4+ and CD8+ cells with significantly higher ratios than to those in PBS control or S-iso-IT-treated animals. The relatively lower fractions of saporin-bound cells in IVS− pools may reflect the slower diffusion rates of S-CD3e-IT to tissue parenchyma compared to those of smaller mAbs. The binding of S-CD3e-IT to other cell types, including CD19+ B cells and granulocytes, was negligible in both IVS+ and IVS− pools (Figure 4B). S-iso-IT did not show any measurable binding to any leukocytes in either the IVS+ or IVS− pools (Figure 4B).

We then evaluated the depletion of tissue-resident T cells using the same treatment and intravascular staining conditions described above (see Figure 3A). Consistent with the in vivo binding assay, S-iso-IT did not show any notable reduction of T cells (Figure 5A). B cells were notably reduced in the peripheral blood, but also showed a moderate increase in the IVS− pool of other organs, indicating the potential migration of these cells. However, S-CD3e-IT effectively and specifically depleted T cells both in the peripheral blood and in IVS− tissue sites (Figure 5A). T-cell depletion was most efficient for IVS+ circulating peripheral blood T cells (average 96.5%) and the depletion rates averaged 89.3% and 79.7%, and 84.7% for tissue resident (IVS−) T cells in the spleen, lungs, and LNs, respectively, indicating the effective depletion of the majority of tissue-resident T cells in these organs. CD3e MFI analysis on IVS− CD4+ and IVS− CD8+ cells showed similar CD3 modulation and cell depletion patterns to those of the peripheral blood (Figure 5C). Similar to the total CD4+ and CD8+ cell analysis (shown in Figure 2), S-CD3e-mAb treatment nearly completely modulated CD3e expression on T cells in the tissue parenchyma, while S-CD3e-IT depleted them (Figure 5C).

It is notable that, despite the negligible cell death by S-iso-IT in the spleen and lungs, S-iso-IT-treated animals showed significant signs of splenomegaly and capillary leakage (see Figure 2B and Figure 3C). These results are consistent with the previous reports following an administration of Ricin toxin A chain in rats [85]. Interestingly, S-iso-IT showed stronger signs of capillary leakage compared to S-CD3e-IT (Figure 3C). This difference may also reflect the difference in the cell internalization ratios between S-CD3-IT and S-iso-IT, allowing a higher number of S-iso-ITs to circulate and perhaps interact with endothelial cells to cause vascular leakage. Similarly, a previous study showed that a truncated DT (DT390) alone was more toxic than CD3-DT390 (DT390anti-CD3sFv) in inducing renal toxicity in mice [86]. Taken together, these results demonstrate that S-CD3e-IT effectively depletes T cells both in circulation and local tissues. Nevertheless, S-iso-IT-mediated capillary leakage indicates that the impact of local T-cell death itself is not substantial in inducing VLS.

### 3.7. Direct Comparison of S-CD3e-IT and Diphtheria Toxin Treatment in Mice

Lastly, we compared S-CD3e-IT and DT treatments in mice. VLS has been well documented in DT-based IT studies [44,87], but not with saporin-ITs. A DT-conjugated to S-CD3e-mAb would have been an ideal comparison for this purpose, but previous murine-version CD3e DT has shown lethal renal toxicity [86]. As an alternative, we tested DT and S-CD3e-IT in transgenic CD4-iDTR mice. CD4-iDTR mice express DT receptors on the surface of CD3+ T cells (both CD4+ and CD8+ T cells due to the CD4 promoter-driven DTR expression) [58] and an injection of DT was shown to effectively and specifically deplete T cells similarly to S-CD3e-IT (Figure 3A, left). These mice, therefore, provide a unique opportunity to directly compare DT and S-CD3e-IT for T-cell depletion and vascular leakage in various organs. As expected, both DT and S-CD3e-IT effectively and specifically depleted both IVS+ and IVS− T cells in all tested organs, including the peripheral blood, spleen, lymph nodes, bone marrow, and lungs (Figure 6B). The DT groups showed significantly higher levels of IVS+ and IVS− PMNs in the spleen and lungs compared to the PBS control group (* *p* < 0.05 and ** *p* < 0.01) (Figure 6C). The W/N ratios also significantly increased in the spleen (* *p* < 0.05) and marginally in the lungs compared to those of the PBS group (Figure 6D). S-CD3e-IT treatment showed comparably high PMN mobilization and infiltration in these organs. These results, thus, further confirm that the saporin–streptavidin (sZAP) molecule of S-CD3e-IT plays a major role in inducing capillary leakage in mice.

## 4. Discussion

In vivo toxicity and pharmacodynamics of CD3e ITs have been evaluated in humans, monkeys, and pigs using species-specific CD3e ITs, including Resimmune (A-dmDT390-bisFv(UCHT1)) [34] for humans, A-dmDT390-scfbDb(C207) for monkeys [33], and A-dmDT390-biscFv(2–6–15) for pigs [88]. Despite the numerous advantages of using mouse models, including a tremendous number of analytic resources and recent technological advances in tracking tissue-resident leukocytes, murine models have been used only rarely for CD3e IT studies since the late 1990s, primarily due to the lack of appropriate anti-murine CD3e ITs—that is, CD3e ITs with safety and treatment efficacy profiles comparable to those of the “second-generation” recombinant CD3e ITs developed for humans, monkeys, and pigs. This study demonstrates the development of new murine-version CD3e IT (S-CD3e-IT) and a new VLS study model that enables a sensitive evaluation of leukocyte circulation and tissue infiltration in mice.

Using a new murine VLS study model, we demonstrated the distinct PMN migration patterns of non-mitogenic 145-2C11 (S-CD3e-mAb) and S-CD3e-IT in mice. It has been presumed that both the CD3e-binding portion (CD3e mAbs) and the toxin portion of CD3-IT can induce VLS without a clear understanding of the underlying molecular mechanisms. In particular, due in part to the severe side effects of first-generation OKT3 and FN18-CRM9 (containing foreign immunoglobulin) reported in previous studies, T-cell activation by the CD3e-binding portion of CD3 IT was thought to induce VLS [20,21,22,23,24,25]. Our murine VLS model studies clearly demonstrate that the toxin molecule (saporin) itself can induce vascular leakage. In our model, the CD3e-binding portion of S-CD3e-IT (S-CD3e-mAb) does not play a major role in inducing vascular leakage. In addition, an isotype antibody-conjugated IT (S-iso-IT) was capable of increasing PMN infiltration without T-cell killing.

The saporin portion of sZAP, rather than the streptavidin portion, is most likely responsible for vascular leakage. Streptavidin or avidin has long been tested in humans and animals as a carrier for therapeutics or diagnostic purposes without notable sign of toxicity [89]. The natural form of these molecules has been shown to be quickly cleared from the blood (with minutes to a few hours of half-life) [90]. Anti-avidin antibodies are common in the general human population due to an exposure to avidin via diet or common vaccines [91,92], but the impact of the immunogenicity of avidin has shown to be negligible on the safety and efficacy of avidin-mediated therapy in humans and mice [93].

The non-mitogenic CD3e-mAb (S-CD3e-mAb) used in this study is a recombinant 145-2C11 with Fc-silent murine IgG1, which mimics the clinical anti-human OKT3 antibodies with Fc-silenced human IgG1. This form of non-mitogenic 145-2C11 has not yet been tested for in vivo pharmacodynamics. Surprisingly, despite the significant VLS induction by wild-type 145-2C11 (with hamster IgG1), S-CD3e-mAb did not show any notable sign of vascular leakage, splenomegaly, or body weight loss, while demonstrating expected in vivo efficacy.

These results point to the importance of the technical advances in antibody and IT engineering that simplify and optimize protein agents by, for example, removing unwanted portions of antibodies or toxins [29,30,31,32]. Indeed, despite the complex potential molecular mechanisms underlying VLS, the engineering of VLS-inducing proteins—by point mutations targeting the (*x*)D(*y*) motif of ricin toxin A chain, DT, or pseudomonas exotoxin A, for example—has been shown to substantially reduce VLS, while maintaining the catalytic activities of these toxic proteins [37,38,39,40,44,94,95]. Although the second-generation recombinant CD3-IT has shown much improved safety profiles, our data support the understanding that the toxin portion of IT can be a major source of side effects.

Saporin is a potent and highly useful cell-killing agent for the development of IT therapies. Useful features of saporin include low cytotoxicity when not internalized (it lacks a cell-binding domain), strong cell-killing activity once internalized, molecular stability, resistance to chemical procedures, and the potential to produce saporin-ITs in yeast [52,53]. Consistent with our study, a previous study demonstrated that CD3-saporin is highly efficient in inducing cell death [96]. When directly compared with other ribosomal inactivating proteins and antibody combinations, CD3-saporin was faster and strongest in inducing cell death [96]. However, the ability of saporin to induce VLS remains poorly characterized. There has been no clear in vitro and in vivo evaluation, and previous clinical studies remain somewhat controversial: clinical trials with saporin-ITs in 1990s reported no clear sign of vascular leakage [54,55], except for one CD30-saporin clinical trial reporting one case of edema out of four patients [56]. We demonstrate here that both S-iso-IT and S-CD3e-IT were highly potent in inducing VLS in mice. Saporin has two known (*x*)D(*y*) motifs, one of which is well exposed to the external surface of the molecule, indicating its potential ability to bind to integrins and damage endothelial cells [44,52]. The (*x*)D(*y*) motifs of saporin remain uncharacterized in vivo and in vitro.

As a serious adverse effect, VLS has limited the use of IT therapy in clinic. The administration of steroids has been useful in improving the tolerability of ITs [97,98,99]. Advances in antibody engineering and recombinant ITs have, likewise, been effective in improving the safety profiles of antibody and immunotoxin therapies. Our study presents experimental data that support the importance of engineering VLS-inducing proteins to generate safer ITs. The new murine testing system we present here can also be a useful option for the in vivo characterization of VLS to enable the safe and widespread use of ITs.

## Figures and Tables

**Figure 1 biomedicines-10-01221-f001:**
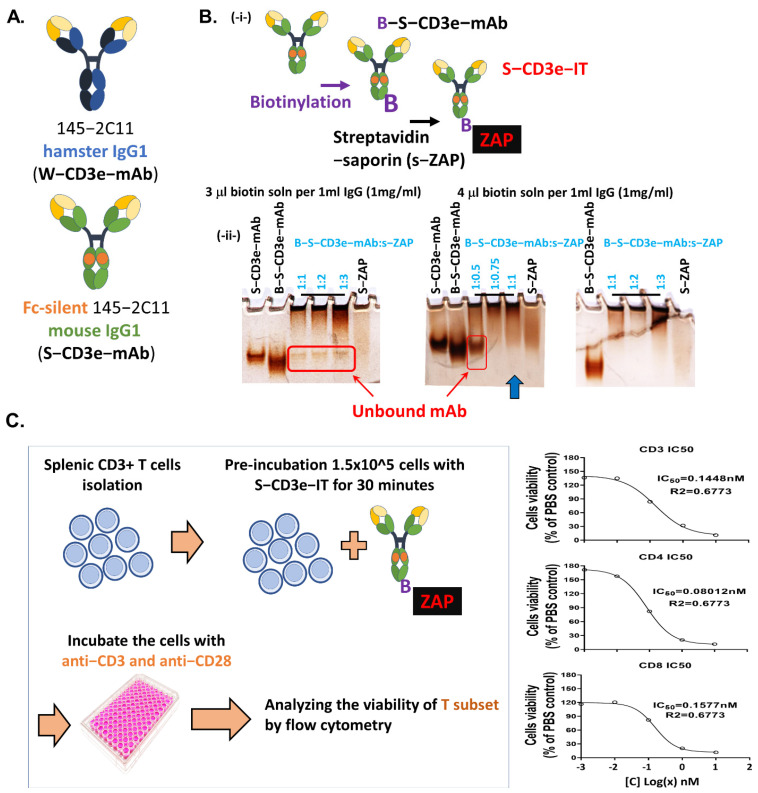
CD3e monoclonal antibodies and new antimurine CD3e immunotoxin. (**A**) AntimurineCD3e monoclonal antibodies. Wildtype hamster 1452C11 (W-CD3e-mAb) or chimeric 1452C11 with Fc-silent mouse IgG1 (S-CD3e-mAb) were tested. (**B**) Generating a new antimurineCD3e immunotoxin (SCD3eIT). (**-i-**) SCD3eIT was generated by first biotinylating SCD3emAbs (BSCD3emAb) and then conjugating BSCD3e-mAb to streptavidin–saporin (sZAP). (**-ii-**) Optimization of the biotinylation step and the antibodysZAP conjugation step. Biotinylation with 4 μL biotin solution (FluoReporter Protein Labeling Kit, Invitrogen) per 1 mL of 1 mg/mL immunoglobulin G (IgG) and conjugation of BSCD3e-IT and sZAP at molar ratio of 1:1 (blue arrow) was an optimal condition to minimize unbound antibodies (red boxes). SCD3emAb, B-S-CD3e-mAb, different ratios of B-S-CD3e-mAb:sZAP, and sZAP are shown on 38% Tris–acetate gradient gels. (**C**) In vitro SCD3eIT assay. Splenic CD3+ T cells were incubated with 1 pM to 10 nM of S-CD3e-IT for 30 min, then incubated in antiCD3 and antiCD28 72 h. Cell viability was assayed by flow cytometry analysis. Cell viability at different concentrations of SCD3eIT was normalized with that of no-treatment (PBS) control.

**Figure 2 biomedicines-10-01221-f002:**
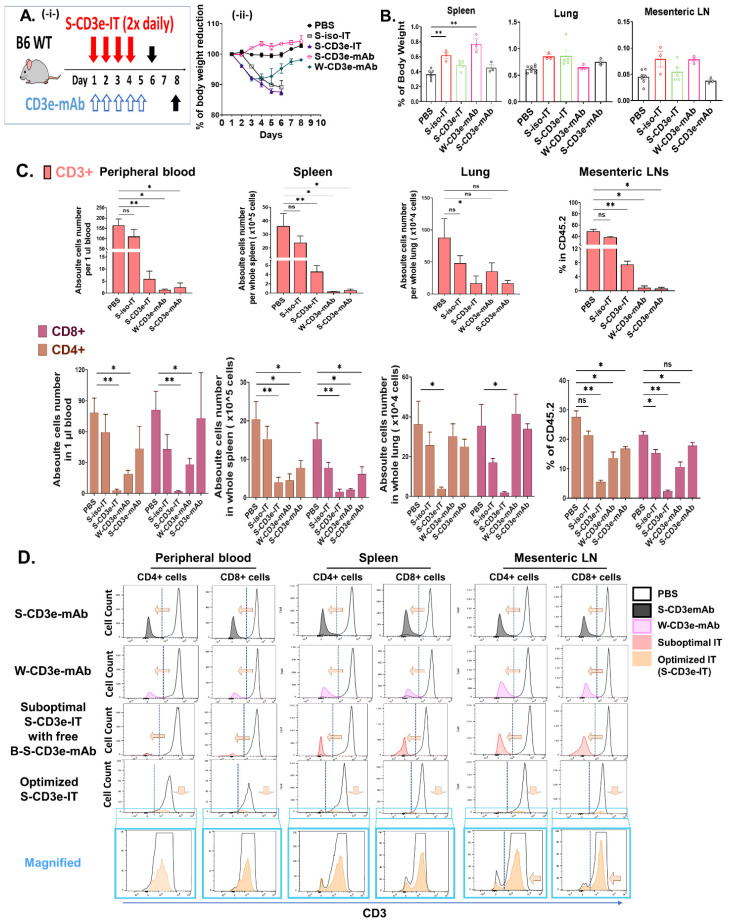
Distinct modes of action by non-mitogenic 145-2C11 and S-CD3e-IT in mice. (**A**) Treatment strategy is shown on the left (**-i-**) and a chart for the body weight changes over time is shown on the right (**-ii-**). (**B**) Organ weights are shown as % of body weight. (**C**) CD3+ cells (**upper panel**) and CD4+ and CD8+ lymphocytes (**lower panel**) are shown for the peripheral blood (absolute number per 1 μL blood), spleen (absolute number per whole spleen), lung (absolute number per whole lung), and mesenteric lymph nodes (LNs; % in CD45.2+ leukocytes). Cell numbers were determined based on CountBright Absolute counting beads and the total cell counts per organ. (**D**) Mean fluorescent intensity (MFI) of CD3e on the CD4+ and CD8+ cells (x-axis) and cell count (y-axis) are shown for the peripheral blood (**left two panels**), spleen (**middle two panels**), and mesenteric LNs (**right two panels**). S-CD3e-mAb nearly completely modulated CD3e surface expression on these cells while maintaining the total T cell numbers; by contrast, S-CD3e-IT showed an effective depletion of T cells with a mild decrease in CD3e MFI on the surviving CD4+ and CD8+ cells. Blue boxes are magnified views of the MFI plot for the optimized S-CD3e-IT. Non-mitogenic 145-2C11 (S-CD3e-mAb; *n* = 3), wild-type 145-2C11 (W-CD3e-mAb; *n* = 3), sub-optimally prepared IT (suboptimal S-CD3e-IT with free B-S-CD3e-mAb; *n* = 3), and optimized S-CD3e-IT (*n* = 5) were compared. (* *p* < 0.05 and ** *p* < 0.01).

**Figure 3 biomedicines-10-01221-f003:**
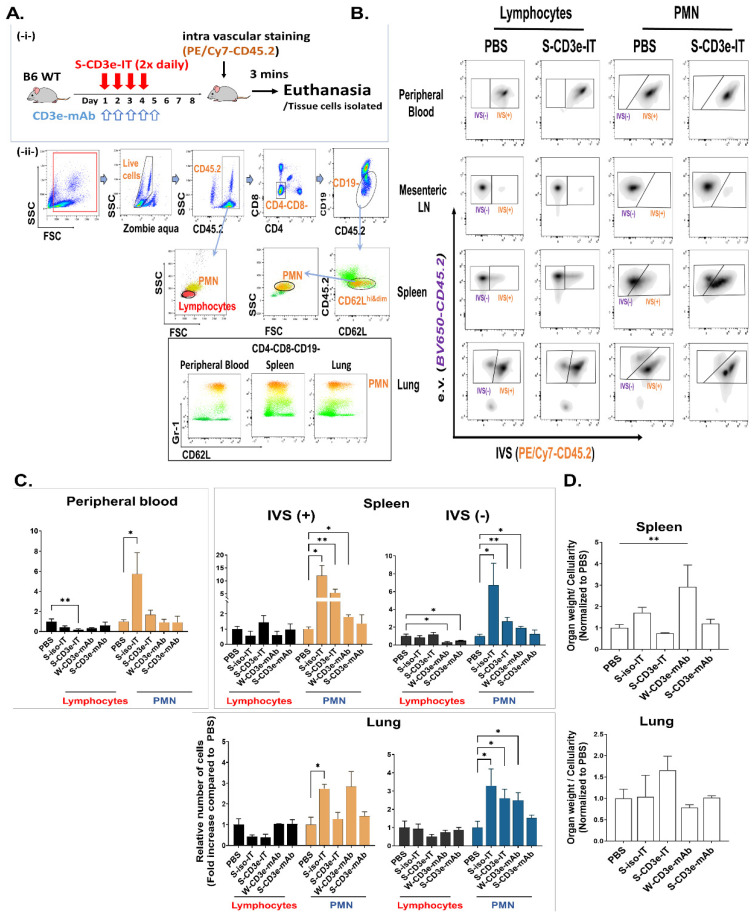
Tracking capillary leakage in mice. (**A**) The upper panel (**-i-**) shows a strategy to track leukocyte migration from the vasculature to the tissue parenchyma, and the lower panel (**-ii-**) shows flowcytometry gating strategy to define polymorphonuclear leukocytes (PMNs) and lymphocytes. The bottom panel shows varying CD62L expression of Gr-1^high^ PMNs from CD4-CD8-CD19 cells in different tissues. (**B**) Defining intravascular staining antibody positive (IVS+) and negative (IVS−) populations for lymphocytes (**left two columns**) and PMNs (**right two columns**). (**C**) Lymphocyte and PMN population changes in the vasculature (IVS+) and in the tissue parenchyma (IVS−). The total numbers of IVS+ or IVS− lymphocytes (black bars) and PMNs (orange bars for IVS+ PMNs and blue bars for IVS− PMNs) per organ were divided by those of the PBS control (y-axis: fold increase compared to PBS) to estimate the relative changes of these cells after different treatments. IVS+ or IVS− lymphocytes and PMNs are shown side-by-side for the peripheral blood, spleen, and lungs separately. (**D**) The ratios of organ weight to total cell count recovered from the organ (W/N ratios) are shown for the spleen (upper panel) and lung (lower panel). (* *p* < 0.05 and ** *p* < 0.01); *n* = 7 for PBS; *n* = 3 for S-iso-IT; *n* = 5 for S-CD3e-IT; *n* = 3 for W-CD3e-mAb; *n* = 3 for S-CD3e-mAb).

**Figure 4 biomedicines-10-01221-f004:**
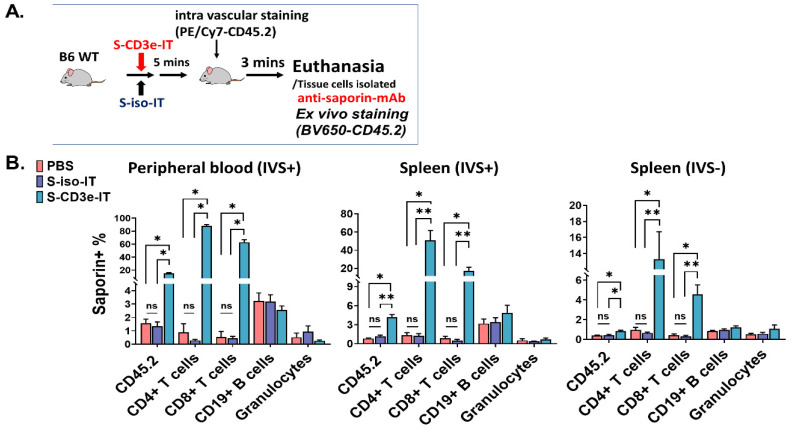
In vivo binding assay reveals that S-CD3e-IT, but not S-iso-IT, binds to CD4+ and CD8+ cells specifically both in the circulating (IVS+) and tissue-resident (IVS−) cell pools of the spleen. (**A**) a strategy to analyze cell binding in both the vasculature and the tissue parenchyma. Mice received a single injection of S-CD3e-IT or S-iso-IT, then five minutes later received IVS (PE/Cy7-CD45.2), and three minutes later were subjected to euthanasia and ex-vivo cell staining. (**B**) Percent of cells with saporin molecules on the cell surface (y-axis) are shown for CD45.2+ cells, CD4+ cells, CD8+ cells, CD19+ B cells, and PMNs, in IVS+ and IVS− pools of the spleen. (* *p* < 0.05 and ** *p* < 0.01; *n* = 4 for PBS, *n* = 4~5 for S-iso-IT; *n* = 4~5 for S-CD3e-IT).

**Figure 5 biomedicines-10-01221-f005:**
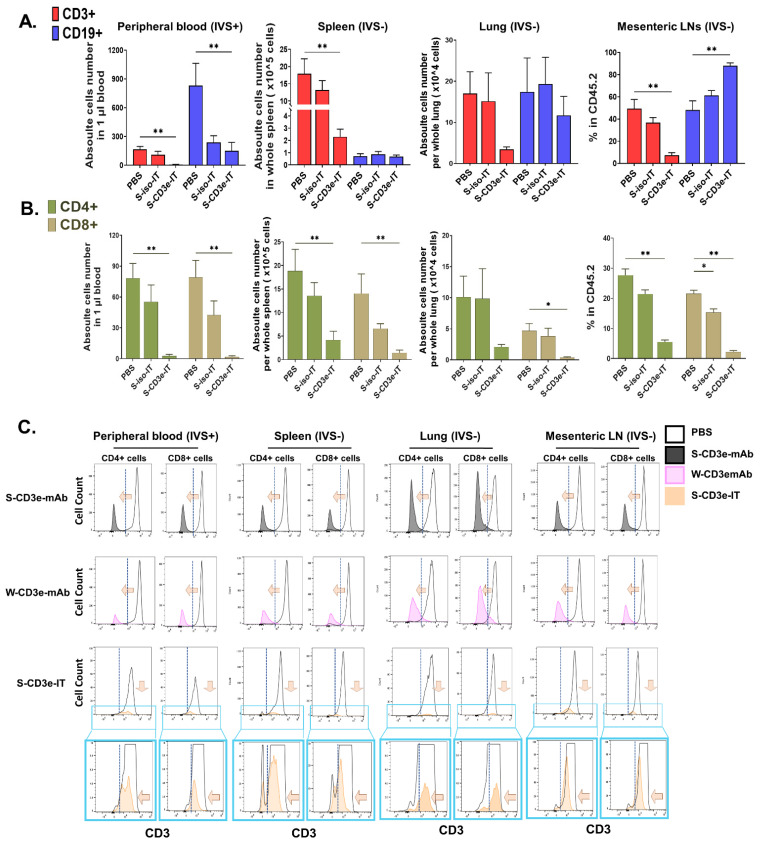
Effective depletion of both circulating and tissue-resident T cells by S-CD3e-IT. (**A**,**B**) CD3+ T cells (**A**) and CD4+ and CD8+ lymphocytes (**B**) are shown for the peripheral blood (absolute number per 1 μL blood), IVS− spleen (absolute number per spleen), IVS− lung (absolute number per lung), and IVS− mesenteric lymph nodes (LNs; % in CD45.2+ leukocytes). (**C**) Mean fluorescent intensity (MFI) of CD3e on the CD4+ and CD8+ cells (x-axis) and cell count (y-axis) are shown for the peripheral blood, IVS− spleen, IVS− lungs, and IVS− mesenteric LNs. Blue boxes are magnified views of the optimized S-CD3e-IT MFI plot. (* *p* < 0.05 and ** *p* < 0.01: *n* = 7 for PBS; *n* = 3 for S-iso-IT; *n* = 5 for S-CD3e-IT).

**Figure 6 biomedicines-10-01221-f006:**
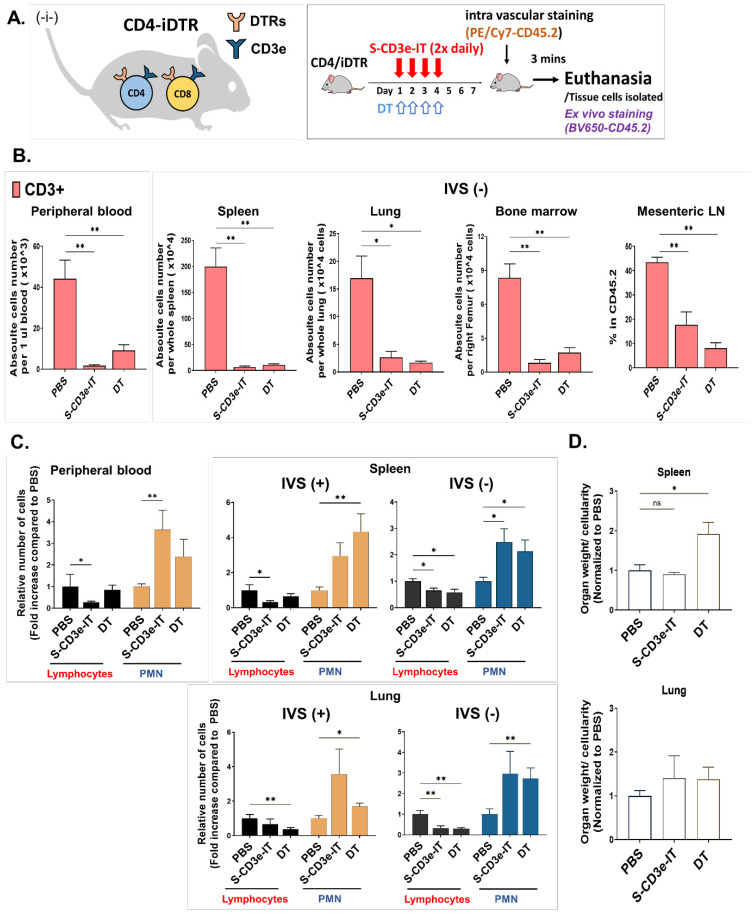
(**A**) CD4iDTR transgenic mice expresses diphtheria toxin receptors (DTRs) on the surface of CD4+ and CD8+ T cells (**-i-**). The right box shows treatment and analytic strategies. Mice were treated with S-CD3e-IT or DT twice daily for four consecutive days and subjected for intra vascular staining and euthanasia on Day 7. (**B**) CD3+ cells are shown for the peripheral blood (absolute number per 1 μL blood), IVS− spleen (absolute number per whole spleen), IVS− lung (absolute number per whole lung), IVS− bone marrow (absolute number per right femur) and mesenteric LNs (% in CD45.2+ leukocytes). (**C**) Lymphocyte and PMN population changes in the vasculature (IVS+) and in the tissue parenchyma (IVS−). The total numbers of IVS+ or IVS− lymphocytes (black bars) and PMNs (orange bars for IVS+ PMNs and blue bars for IVS− PMNs) per organ were divided by those of the PBS control (y-axis: fold increase compared to PBS). IVS+ or IVS− lymphocytes and PMNs are shown side-by-side for the peripheral blood, spleen, and lungs separately. (**D**) The ratios of organ weight to total cell count recovered from the organ (W/N ratios) are shown for the spleen (**upper panel**) and lung (**lower panel**). (* *p* < 0.05 and ** *p* < 0.01; *n* = 8 for PBS, *n* = 11 for DT; *n* = 6~7 for S-CD3e-IT).

## Data Availability

All data related to the study described in the manuscript are described in the report.

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
