# Peer review of "Comparison of CD3e Antibody and CD3e-sZAP Immunotoxin Treatment in Mice Identifies sZAP as the Main Driver of Vascular Leakage"

_biomedicines, 2022, doi:10.3390/biomedicines10061221_

Round 1
Reviewer 1 Report
The topic that the authors have addressed is a significant problem faced during toxin-conjugated mAb therapies. The approach follows the hypothesis and results well explained. The research has high clinical significance and falls within the scope of the journal.
Minor concerns:
- It is interesting that both S-CD3 and S-CD3-IT mAbs have almost same efficacies in reducing T cells (Figure 2). While, as authors hypothesized, modulation of CD3 expression by these antibodies is a valid explanation, it is also possible that the staining anti-CD3 mAb used in flow cytometry might have been competitively inhibited by therapeutic mAb binding? Can the authors comment on such possibility and discuss whether such cross-blocking is known for the clone they used in this study?
- It is understandable that abbreviations need to be used to keep the length of the manuscript within limit. But having too many non-standard abbreviations in this manuscript makes it harder to read. So the authors might consider reducing the usage of non-standard abbreviations, at least.
Author Response
Hi,
Please see the attachment.
Thanks,
Shihyoung

Reviewer 2 Report
The goal of these studies is to demonstrate that saporin, like all other toxins used in immunotoxins (ITs), can cause vascular leak syndrome. To this end the authors have performed an extensive and generally well-designed set of experiments comparing a naked anti-CD3e mAb and an IT made with the same engineered mAb. While I generally believe the conclusions, I have two concerns that need to be addressed:
- The IT was constructed using biotinylated-mAb and avidin-saporin. Thus, when comparing the IT to the naked mAb, saporin is not the only difference between the two. There is also the biotin-avidin complex. How can we be sure that the observed biological effects are solely due to saporin? While I think it likely that the VLS is due to the saporin, I can also imagine plausible mechanisms whereby avidin could mediate effects. Presumably at some point in their studies the authors have coupled a non-toxin (eg albumin) to the mAb via biotin-avidin, or added avidin to the biotinylated mAb, and shown that such constructs do not induce VLS or inflammation greater than the naked mAb.
- The authors have equated PMN infiltration with VLS, best shown in lines 24-26 of the abstract, and this is the sole measure used to assess VLS (except for weights in figure 2). The assay used is excellent, but there are other laboratory parameters that would demonstrate the difference between the induction of inflammation and VLS, which their assay does not do. One simple measure is serum albumin.
A minor point, references 30 and 68 (and perhaps others) are incomplete.
Author Response

(The authors gave the same response as above.)

Reviewer 3 Report
In the paper entitled "Comparison of CD3e antibody and CD3e-saporin immunotoxin treatment in mice identifies saporin as the main driver of vascular leakage", authors evaluated the roles of the binding portion and the toxic moiety of the S-CD3e-IT in vascular leak syndrome (VLS).
VLS is one of the most described side effects in treatment with immunotoxins, but the mechanism is not yet fully understood. I think that this paper clearly contributes to shedding light on the role of saporin in the induction of VLS.
The paper is very-well-written and the experimental design very robust. In my opinion, the authors did a great job. The introduction is detailed, providing sufficient background and including relevant literature about the topic.
Methods are clearly detailed and results are clearly presented.
The only negative note that I can point out is the quality of the figures, which are very small and hardly visible even by increasing the magnification.
Author Response
Hi,
Please see the attachment.
Thnaks,
Shihyoung

Round 2
Reviewer 2 Report
None